# Optogenetic Approaches for the Spatiotemporal Control of Signal Transduction Pathways

**DOI:** 10.3390/ijms22105300

**Published:** 2021-05-18

**Authors:** Markus M. Kramer, Levin Lataster, Wilfried Weber, Gerald Radziwill

**Affiliations:** 1Faculty of Biology and Signalling Research Centres BIOSS and CIBSS, University of Freiburg, 79104 Freiburg, Germany; markus.kramer@cibss.uni-freiburg.de (M.M.K.); levin.lataster@biologie.uni-freiburg.de (L.L.); wilfried.weber@biologie.uni-freiburg.de (W.W.); 2SGBM—Spemann Graduate School of Biology and Medicine, University of Freiburg, 79104 Freiburg, Germany

**Keywords:** control of function, optogenetics, signal transduction

## Abstract

Biological signals are sensed by their respective receptors and are transduced and processed by a sophisticated intracellular signaling network leading to a signal-specific cellular response. Thereby, the response to the signal depends on the strength, the frequency, and the duration of the stimulus as well as on the subcellular signal progression. Optogenetic tools are based on genetically encoded light-sensing proteins facilitating the precise spatiotemporal control of signal transduction pathways and cell fate decisions in the absence of natural ligands. In this review, we provide an overview of optogenetic approaches connecting light-regulated protein-protein interaction or caging/uncaging events with steering the function of signaling proteins. We briefly discuss the most common optogenetic switches and their mode of action. The main part deals with the engineering and application of optogenetic tools for the control of transmembrane receptors including receptor tyrosine kinases, the T cell receptor and integrins, and their effector proteins. We also address the hallmarks of optogenetics, the spatial and temporal control of signaling events.

## 1. Introduction

The cellular signaling network consists of highly intertwined pathways that control the functional response of the cell. Thus, receptor stimulation does not only result in the activation of a particular pathway but instead typically involves signaling proteins across the network, mediated by signal integrators and cross-talks between the pathways. Therefore, an input signal is not processed in a linear pathway but rather in a network, in which all events are more or less connected and result in a combined network output. Furthermore, signaling processes are highly dynamic and the duration and subcellular localization of the activation can determine whether it is passed on or halted. Chemical approaches to control intracellular signaling have been successfully applied [1]. However, they are limited in their spatial and temporal resolution.

Using molecular optogenetics, genetically encoded photoreceptors connected with signaling proteins can orthogonally control the activity of the signaling protein by light. This approach has a number of advantages [2,3]. For example, the activity of the signaling protein can be controlled by the intensity and duration of illumination [4]. Multichromatic approaches enable the control of two signaling pathways independently by using light of different wavelengths [5]. The unprecedented spatiotemporal resolution allows researchers to activate signaling in single cell resolution or even subcellular resolution dynamically [6,7].

Here, we will provide detailed information on the control of intracellular signaling in mammalian cells by opsin-independent molecular optogenetics [8]. Several recent reviews cover the development and application of opsin-based optogenetic tools in neuroscience including the light control of G-protein coupled receptors [9,10,11,12,13,14]. In this review, we first will answer the following questions. How do molecular optogenetic switches function? What are the most commonly used optogenetic switches for the control of intracellular signaling in mammalian cells? Second, we will provide an extensive overview of the optogenetic control of intracellular signaling. How can receptor tyrosine kinases and particular signaling molecules in their downstream pathways be activated by using light? What knowledge has been gained by doing so? Likewise, we will describe T cell receptor (TCR) and integrin signaling. What advantages does optogenetics offer? We will describe, how the highest spatiotemporal precision in the control of signaling proteins led to novel insights into the signaling network. Finally, we will discuss future trends in the field of optogenetics.

## 2. Optogenetic Switches as Basis for Light Control

In the following, detailed information is provided on how optogenetic switches function and which optogenetic switches are available to control intracellular signaling in mammalian cells.

Optogenetic switches consist at least of a photoreceptor that undergoes a conformational change upon illumination with a specific wavelength. Depending on the switch, different modes of action are possible. Optogenetic switches can interact with themselves, thereby forming either homodimers or homo-oligomers. Those are suited as fusion partners for proteins of interest that are active as dimer or oligomer, such as most of the receptor tyrosine kinases. Other switches engage an interaction partner (heterodimerization) which allows artificial split proteins such as in the case of split transcription factors [15] or to target a protein of interest to specific compartments of the cell such as the plasma membrane or the outer mitochondrial membrane by using appropriate membrane anchors [16]. In further switches, exposure to light can induce an intramolecular conformational change that leads to the uncaging of a motif of interest, such that the nuclear import or export can be controlled [17] (Figure 1).

Optogenetic switches are currently available across the visible spectrum and beyond. Optobase.org is a database including a curated up-to-date list of available opto-switches developed so far [8]. The most commonly used switches for controlling signaling pathways in mammalian cells are the red light-inducible phytochromes as well as the blue light-inducible cryptochromes and light-oxygen-voltage (LOV) domains (Table 1).

Phytochromes form a large group of optogenetic switches derived from bacteria and plants. The most commonly used phytochrome photoreceptor is phytochrome B (PHYB) derived from *Arabidopsis thaliana*, which uses the cofactor phycocyanobilin that is not endogenous to mammalian cells [18]. Therefore, it has to be added externally or synthesized in situ by the co-expression of genes encoding biosynthetic enzymes [35,36]. Upon 660 nm red light illumination, PHYB undergoes a conformational change within milliseconds and forms heterodimers with the phytochrome interacting factor (PIF) (Figure 1A). This interaction is stable in the dark for hours or can be terminated by 740 nm far-red light within milliseconds [18]. Another switch is CPH1, which is based on the photoreceptor cyanobacterial phytochrome CPH1 derived from *Synechocystis* sp. PCC 6803 [19]. CPH1 forms homodimers upon red light exposure, with a reversion time in the millisecond-scale in the dark and upon far-red light illumination (Figure 1B) [19].

All optogenetic switches in the class of cryptochromes are based on the photoreceptor cryptochrome 2 (CRY2) derived from *Arabidopsis thaliana*. CRY2 incorporates the cofactor flavin adenine dinucleotide (FAD), resulting in an absorption maximum of around 450 nm. Upon exposure to blue light, CRY2 undergoes a conformational change that enables it to bind to its interaction partner cryptochrome-interacting basic helix-loop-helix 1 (CIB1) and forms heterodimers (Figure 1C) [20]. In most cases, this switch is optimized by using a truncated version of CRY2 comprising the CRY2-photolyase homology region (CRY2-PHR, named in this review CRY2) and the N-terminal amino acids of CIB1 (CIBN, aa 1-100). Additionally, CRY2 can form homo-oligomers (Figure 1D) [21]. The charge of the C-terminus of CRY2 affects the strength of cluster formation. Positive charges enhance CRY2 homo-oligomerization (CRY2olig, CRY2clust, and CRY2high [25,26,27]), whereas negative charges reduce it (Cry2low [27]). CRY2 switches typically have an excitation time in the range of seconds and a dark reversion within minutes [20].

LOV domain-containing proteins are reversible light sensors found in bacteria, fungi, and plants. All LOV domains bind flavin mononucleotide (FMN) or flavin adenine dinucleotide (FAD) as a cofactor, resulting in an absorption maximum around 450 nm [22,23,37]. However, a wide variety of different LOV domains and modes of action is available. Upon blue light illumination, the LOV domain of aureochrome 1 (AuLOV) from the alga *Vaucheria frigida* (VfAU1-LOV) forms homodimers [31]. Magnet is a pair of heterodimers derived from the homodimer forming blue light switch Vivid (VVD) from the fungus *Neurospora crassa*. The Magnets pair, pMag, and nMag, interact with each other through electrostatic interaction and can perform switch-off kinetics in the range of seconds up to many hours, depending on the variant [29].

Another type of LOV domain used in intracellular signaling is based on the LOV2 domain of phototropin 1 derived from *Avena sativa* (AsLOV2) (Figure 1E). Blue light illumination induces an intramolecular conformational change that leads to the unwinding of the Jα-helix into an open structure [38]. A motif of interest can be fused C-terminally to the Jα-helix, such as a nuclear localization or nuclear export signal to shield it in the dark and uncage it under blue light creating the LINuS and LEXY systems [17,39]. In the switch improved light-induced dimer (iLID) the bacterial peptide SsrA has been embedded in the Jα-helix, enabling light-induced dimerization with its binding partner SspB (Figure 1F) [24]. In the switch LOVTRAP, the binding partner Zdk engages the Jα-helix directly in the dark and dissociates by illumination [28]. AsLOV2-based switches have an excitation time in the range of seconds and typically a reversion time of tens of seconds to minutes. Additional variants are available for LOV-based switches [24,29,30].

## 3. Optogenetic Control of Receptor Tyrosine Kinase Signaling

Receptor protein kinases (RTK) are transmembrane proteins with intrinsic tyrosine-protein kinase activity [40]. Most RTKs are inactive monomers or preformed inactive dimers in the unstimulated state. Upon ligand binding, functional dimers or oligomers are formed (Figure 2A). Activation of RTKs leads to transphosphorylation of their tyrosine kinase domain and recruitment of effector proteins containing phosphotyrosine binding domains [41]. Pathways stimulated by RTKs include the phosphatidylinositol-3-kinase (PI3K)/AKT (also named protein kinase B) pathway and the mitogen-activated protein kinase (MAPK) pathway composed of the three protein kinases RAF, MAPK/ERK kinase (MEK), and the extracellular signal-regulated protein kinase (ERK). RTK signaling regulates proliferation, differentiation, survival, and apoptosis, while its dysregulation is linked to human diseases including cancer [42]. Chemical dimerizers have been successfully applied for control of RTK activation. Optogenetics expands the toolbox. The fusion of RTKs to an optogenetic switch enables light-dependent dimerization and hence the spatiotemporal control of RTK signaling (Figure 2) [43].

### 3.1. Light Control of Receptor Tyrosine Kinases

The first light-controllable RTKs developed were the three members of the tropomyosin-related kinase (TRK) family, TrkA, TrkB, and TrkC [44]. Fusion of CRY2 C-terminally to the full-length receptor created a light-responsive chimeric Trk, inactive in the dark and stimulated by blue light-induced clustering (Figure 2B). OptoTrkB regulated MAPK/ERK, AKT, and Ca^2+^ signaling pathways independent of its ligand brain-derived growth factor (BNDF) in a time-reversible, spatial, and cell type-specific manner. Prolonged patterned illumination caused sustained ERK activation inducing neuronal differentiation. The first light-inducible fibroblast growth factor 1 receptor (FGFR1), optoFGFR1, was based on the same optogenetic switch [45]. To prevent endogenous ligand binding and to optimize light control, this chimeric receptor lacked the extracellular domain and the transmembrane region. Instead, a myristoylation signal for plasma membrane-anchoring was placed N-terminally to the cytoplasmic kinase domain and the CRY2 was placed C-terminal to it (Figure 2C). Light exposure transiently activated the canonical pathways phospholipase C γ (PLCγ), PI3K, and MAPK/ERK-dependent on the frequency and duration of the light duration.

In contrast to other RTKs, the erythropoietin-producing hepatocellular (Eph) receptor requires high-order cluster formation. In the case of the receptor EphB2, fusion of the intracellular domain to a myristoylation sequence and CRY2 wild-type was not functional. However, variants fused with the mutant CRY2olig (CRY2-E490G) [25] showing a higher tendency for cluster formation induced EphB2 signaling [46]. Light-stimulated EphB2 drove dendritic filopodia formation depending on the activation of the Abelson tyrosine-protein kinase 2 but without activating the GTPases Rac1 and CDC42 often involved in the reorganization of the cytoskeleton. Expression of optoEphB2 in the lateral amygdala of mice increased activation of the tyrosine kinase SRC and cAMP/Ca^2+^-responsive element binding (CREB) protein correlating with enhanced long-term memory [47].

To create additional optogenetic switches, a panel of different blue light-sensing LOV domains was analyzed for optical control of FGFR1 and the epidermal growth factor receptor (EGFR) [31]. The dimer-forming LOV domain of the alga *Vaucheria frigida* aureochrome1, named VfAU1-LOV or AuLOV, C-terminally fused to the intracellular domain equipped with a myristoylation sequence at the N-terminus proved to be the best variant. Light stimulation of optoFGFR1(AuLOV) was coupled to ERK, AKT, and PLCγ1 signaling with temporal precision on the minute time scale. Expression of optoFGFR1 in endothelial cells induced sprouting in spheroids to a similar extent as the major proangiogenic factor VEGF-A. Thus, optoRTKs could functionally substitute signaling by ligands and control the behavior of cells. Based on this approach, an all-optical platform in a 384-well-plate format was developed for the screening of small molecules against protein kinases [48]. Blue light served as a universal stimulus for receptors including FGFR, EGFR, and ROS1 and an ERK pathway-responsive GFP reporter as readout. OptoFGFR1-(AuLOV) confirmed the effect of FGFR1 signaling on differentiation and neurite outgrowth in PC12 cells [49]. Differently localized optoFGFR1 variants proved that induction of neuronal differentiation depends on the association of optoFGFR1 to the plasma membrane as cytoplasmic or nuclear variants are not functional.

To expand the light spectrum for control of RTKs, a codon-optimized variant of the sensory module of CPH1(CPH1S-o) was applied forming monomers in the dark and dimers upon 630 nm exposure [50]. FGFR1-CPH1S-o and TrkB-CPH1S-o, composed of the full-length RTK fused to the photosensor domain, directly activated ERK signaling in cells and transdermally in tissue with 630 nm light demonstrating the advantage of deep-tissue penetration by long-wavelength red light. For multichromatic experiments tools sensing different wavelengths are necessary. The same group developed a green light-sensitive FGFR1 by fusing its intracellular domain to the cobalamin (vitamin B12) binding domain (CBD) of bacterial CarH transcription factors derived from *Myxococcus xanthus* (MxCBD) and *Thermus thermophiles* (TtCBD) [33]. Dimers of CDB form tetramers in the dark and dissociate to monomers with green light. Thus, the myristoylated intracellular kinase domain of FGFR1 fused to CBD formed plasma membrane-bound oligomers in the dark correlating with continuously stimulated FGFR1 signaling. Otherwise, green light disrupted the oligomers and shut FGFR1 signaling off. FGFR1-CBD was used to manipulate zebrafish embryogenesis.

To optimize the control of RTK signaling, several approaches divide the activation of RTKs into two steps: dimerization of the intracellular RTK domain in the cytoplasm and recruitment of the complex to the plasma membrane (Figure 2D). Based on the CRY2/CIBN switch, CRY2 fused of the intracellular domain of TrkA (iTrkA) mediated cytosolic oligomerization and membrane-anchored CIBN-CaaX recruited the oligomeric complexes to the plasma membrane [51]. While cytosolic CRY2-iTrkA oligomers were sufficient for stimulation of ERK and AKT signaling upon light illumination, coupling with plasma membrane recruitment by CIBN1-CaaX more effectively mimicked the role of the nerve growth factor (NGF) in inducing neurite outgrowth of PC12 cells and survival of neuronal cells. To facilitate dimer formation at the plasma membrane, plasma membrane-anchored CIBN coupled to dimers can be used. In this cytoplasm to membrane (CMT) approach, the cytoplasmic domain of FGFR1 fused to CRY2 was coupled with recruitment to CIBN dimers equipped with the CaaX sequence [52]. Plasma membrane-anchored CIBN dimers may not be essential for this approach as FGFR-CRY2 already forms oligomers. However, steric and kinetic effects may positively affect the activation of optoRTK. CMT-optoFGFR as well as CMT-TrkA and CMT-TrkB showed minimal toxicity and induced ectopic tail-like structures in *X. laevis* embryos. This strategy resulted in optical activation of RTKs with low background activity and high sensitivity. The iLID optoTrkA system is another variant of the two-step approach [53]. iLID is based on the release of the SsrA peptide upon light exposure allowing the binding to its interaction partner SspB. Fusions of iTrkA with iLID were cytosolic monomers. Myristoylated tandem dimers of SspB enabled light-dependent dimerization of iLID-TrkA at the plasma membrane correlating with stimulation of TrkA downstream signaling. This system is more accurate, as monomeric iLID-TrkA is inactive in contrast to CRY2-TrkA oligomers.

For regulation with near-infrared (780 nm) and far-red (660 nm) light, the kinase domain of TrkA and TrkB were fused to the C-terminus of the myristoylated photosensory core module of the *Deinococcus radiodurans* bacterial phytochrome (DrBphP) [34]. Under 780 nm light or in darkness, DrBphP adopted a Pr state leading to the formation of closed dimers coupled with activation of Trk (Dr-Trk). Exposure to 660 nm light resulted in the open Pfr state, in which the DrBphP fusion proteins dissociated apart which correlated with the inactivation of Trk (Figure 2E). Thus, 660 nm light-triggered inactivation of Trk when fused to DrBphP and activation in case of CPH1 as mentioned above. Injection of plasmids coding for Dr-TrkA in mammary glands of female mice enabled the activation of TrkA signaling in mice illuminated with 780 nm light. For a two-step control, Dr-TrkA was combined with the TULIP switch. The cytoplasmic version of Dr-TrkA was fused to ePDZ and AsLOV2 carrying the respective PDZ ligand motif was anchored to the plasma membrane. 780 nm light led to a weak stimulation of cytoplasmic Dr-TrkA by dimer formation. However, the combination of 447 nm and 780 nm targeted cytoplasmic Dr-TrkA dimers to the plasma membrane and fully activated TrkA signaling. Recently, the DrBphP system was also applied for control of EGFR and FGFR1. [54]. The respective optoRTKs efficiently triggered ERK1/2, PI3K/AKT, and PLCγ signaling. DrBphP-RTKs may be an interesting tool for spectral multiplexing with blue light-regulated signaling.

To delineate the signaling outcome of a specific docking site for effector proteins, the role of key physiological tyrosine residues in the RTK kinase domain were studied with TrkA-AuLOV [55]. In the case of TrkA, phosphorylation of Tyr 490 promoted the RAF/MEK/ERK signaling pathway, whereas phosphorylated Tyr 785 associated with PLCγ and stimulated PLCγ downstream signaling. Accordingly, the mutant optoTrkA-Y490F (exchange of Tyr 490 to Phe) did not promote PC12 cell differentiation depending on ERK signaling. The mutant optoTrkA-Y785F (exchange of Tyr 785 to Phe) proved that Tyr 785 was essential for PLCγ activity but still gave input to ERK signaling. Both mutants significantly reduced light-induced differentiation indicating that each axis contributed to PC12 differentiation. Thus, PLCγ can feed via protein kinase C (PKC) in RAF/MEK/ERK pathway.

To reveal mechanistic insights in TGFβ-signaling, optoTGFBR was engineered [7]. The natural ligand TGFβ acts as a dimer and binds to the homodimeric, constitutively active TGFβ receptor type II (TβRII). The ligand-bound TβRII dimer enables the formation of a complex with a TGFβ receptor type I (TβRI) homodimer correlating with its transphosphorylation and activation. OptoTGFBR bases on the light-regulated interaction between the cytoplasmic domain of TβRI inserted between the myristoylation signal and CIBN and the cytoplasmic domain of TβRII fused to CRY2. OptoTGFBR allowed the spatiotemporal control of TGFβ signaling in response to different patterns of blue light and enabled to study of the dynamics of SMAD signal complexes, the effector proteins of TβRI, and SMAD-mediated gene expression.

Another optogenetic approach aimed at light-inducible manipulation of endogenous RTKs activity was termed Clustering Indirectly using Crytrochrome 2 (CLICR, Figure 2F) [56]. CLICR-induced signal activation is based on a switch between a monomeric cytoplasmic binding domain of an adaptor fused to CRY2 and light-induced clusters with strongly increased avidity to the target receptor. CRY2 fused to the N-terminal Src-homology region 2 (SH2) of PLCγ possessed only a low affinity to its target PDGFRβ as a monomer. However, illumination induced a local enrichment of the low-affinity SH2 domain elevating the avidity to endogenous PDGFRβ correlating with receptor clustering and activation as well as the generation of focal adhesion structures. This modular platform allows the light control of endogenous transmembrane receptors dependent on the specificity of an adaptor domain fused to CRY2 to the respective interaction motif on the target protein.

### 3.2. Optogenetic Control of the RAF/MEK/ERK Axis

Two of the main downstream pathways of RTKs are the PI3K/AKT and RAF/MEK/ERK pathways. In the case of the latter one, RTK recruits the cytoplasmic complex composed of the adaptor protein growth factor-regulated binding protein 2 (GRB2) and the guanine nucleotide exchange factor son-of-sevenless (SOS) to the plasma membrane where SOS catalyzes the generation of active GTP-bound RAS. The main effector of GTP-RAS is the protein kinase RAF that headed the three-tiered protein kinase cascade formed by RAF, MEK, and ERK. Meanwhile, several optogenetic switches are available to control SOS, RAF, and MEK (Figure 3).

The two-component system optoSOS comprises the catalytic domain of SOS (SOScat) fused to PIF and PHYB anchored to the plasma membrane by the prenylation sequence CaaX [57]. Upon red light (660 nm) exposure, PHYB-CaaX recruited PIF-SOScat to the plasma membrane where it catalyzed the exchange of RAS-GDP to RAS-GTP. In turn, active RAS-GTP stimulated the RAF-MEK-ERK protein kinase cascade. Far-red light (740 nm) dissociated the PHYB/PIF complex and shut optoSOS activity off. In contrast to RTKs that activated several downstream pathways, optoSOS selectively activated RAS-mediated ERK signaling. Depending on the light dose and frequency, fast and sustained responding proteins could be activated by optoSOS. In NIH/3T3 fibroblasts, optoSOS induced proliferation comparable to PDGF, whereas in PC12 cells differentiation monitored by neurite outgrowth was induced similar to NGF does. Thus, optoSOS was sufficient to control cell fate decisions by light [57]. In a further study, this optoSOS system was applied to study the dynamics of RAS/ERK signaling underlying the transcription of immediate early genes and the combinatorial control of their protein level [69]. Because the injection of phycocyanobilin, the cofactor essential for PHYB function, is challenging in in vivo experiments, a blue light-responsive optoSOS system was engineered depending on FAD available endogenously [58]. In this case, optoSOS comprised iLID-CaaX as a light switchable membrane anchor and cytoplasmic SspB-SOScat that recruited to it by blue light. In *Drosophila*, the early embryogenesis was sensitive to the spatial distribution of optoSOS-induced ERK signaling disrupting the normal course of morphogenesis. A follow-up study demonstrated that cell fate decision between endoderm and ectoderm depends on the dynamics of ERK signaling in the early *Drosophila* embryo [70].

RAF is the main effector of RAS. Upon growth factor stimulation, RAS-GTP forms nanoclusters facilitating the recruitment of cytoplasmic RAF to the plasma membrane correlating with RAF activation. OptoCRAF, the first light-regulatable protein kinase ever engineered, is based on the light-dependent clustering of CRAF by its fusion to CRY2 [59]. OptoCRAF was sufficient for reversible activation of the RAF-MEK-ERK cascade. The combination of kinase defective BRAF-CRY2 with CRAF-CRY2 or CRAF-CIBN demonstrated the paradoxical activation of CRAF, similar to drug-inhibited BRAF stimulated CRAF. A follow-up study applied this optoRAF toolbox to characterize the effects of clinically approved RAF inhibitors on various homo- and heteromer combinations of CRAF and BRAF [71].

CRAF-CRY2 cluster formation correlates with the stimulation of RAF signaling. Accordingly, CRAF-CRY2high most efficiently stimulated ERK signaling, and CRAF-CRY2 was more active than CRY2-CRY2low [27]. While clustering of RAF-CRY2 mimics stimulation of endogenous RAF by RAS nanoclusters induced by growth factors, co-expression with CIBN-CaaX facilitates light-dependent recruitment of RAF-CRY2 to the plasma membrane. This system successfully induced ERK-dependent neurite outgrowth in PC12 cells [60]. A bicistronic construct composed of CRY2-RAF, a self-cleaving P2A sequence, and a tandem CIBN with a C-terminal CaaX sequence minimized this system to one plasmid and improved the switch conditions [61].

A special type of optogenetic switch depends on the photodissociable dimeric variant of the fluorescent protein Dronpa (pdDronpa) that dimerizes in violet light (400 nm) and dissociates in cyan light (500 nm). Photoswitchable MEK1 (psMEK1) was generated by caging its constitutively active and minimized kinase domain N-terminal and C-terminal with a pdDronpa domain [32]. Cyan light (500 nm) dissociated the Dronpa domains correlating with MEK activity. Violet light (400 nm) rapidly reversed activation of psMEK1 (400 nm). Using a similar design, photoactivatable CRAF was generated by caging the wild type CRAF kinase domain with pdDronpa and equipment of the fusion protein with a CaaX at the C-terminus sequence. psMEK1 and psCRAF induced ERK-mediated physiological effects in *C. elegans* in a light-dependent manner. To optimize psMEK1 the activating phosphomimetic mutations were combined with additional mutations in the kinase domain leading to increased kinase activity without interfering with the reversibility of the system [62]. This optimized psMEK1 was applied to study ERK signaling in Drosophila and zebrafish embryogenesis.

### 3.3. Optogenetic Control of the PI3K/AKT Axis

The second main axis of RTK downstream signaling comprises PI3K and AKT. Specific tyrosine residues phosphorylated by stimulated RTKs *in trans* recruit the lipid kinase PI3K catalyzing the phosphorylation of phosphatidylinositol-4,5-bisphosphate (PIP2) localized in the plasma membrane to phosphatidylinositol-3,4,5-triphosphate (PIP3). PIP3 and PIP2 are substrates of the inositol 5-phosphatase OCRL. AKT binds through its pleckstrin homology (PH) region to PIP3 leading to AKT activation at the plasma membrane by PIP3-dependent kinase 1(PDK1)-mediated phosphorylation of Thr 308 and by mechanistic target of rapamycin complex 2 (mTORC2)-dependent phosphorylation of Ser 473. For all three, OCRL, PI3K, and AKT optogenetic switches are available.

Blue light-induced recruitment of the 5-phosphatase domain of OCRL was achieved by co-expression of CRY2-5-ptase OCRL and CIBN-CaaX [63]. Dephosphorylation of PIP2 and PIP3 by CRY2-5-ptase OCRL correlated with the arrest of clathrin-mediated endocytosis and loss of membrane ruffling. Otherwise, a light-controlled increase in PI3K activity stimulated membrane ruffling. Blue light optoPIP3 (or optoPI3K) comprises the inter-SH2 (iSH2) region of the p85α regulatory subunit of PI3K fused to CRY2 and CIBN-CaaX. iSH2 tightly associates with the endogenous p110 kDa catalytic subunits leading to blue light-dependent localization of this complex to the plasma membrane and phosphorylation of PIP2 to PIP3. Based on the PHYB/PIF switch, iSH2 fused to PIF recruited the endogenous PI3K catalytic domain to the plasma membrane-anchored PHYB-CaaX with red light [65]. Far-red light reversed this effect. Kinetics dependent on the expression levels of the optogenetic compounds and different light inputs revealed a feedback control and facilitated fine-tuning of the lipid levels in single cells. PIP3 levels locally elevated at neurite tips by the blue light optoPIP3 system correlated with the F-actin-dependent formation of filipodia and lamellipodia but did not induce neurite elongation [64]. PIP3 levels could also be modulated by the blue light-dependent Magnet system [29]. Here, nMagHigh1-CaaX recruited iSH2 pMag(3x) elevating PIP3 levels upon blue light exposure accompanied by actin polymerization and formation of lamellipodia and ruffles.

Activated PI3K generates PIP3 that interacts with the PH domain of AKT thereby recruiting cytoplasmic AKT to the plasma membrane for activation. Translocation of AKT to the plasma membrane independently of the PIP3 level could be achieved by the CRY2/CIBN system. Katsura et al. used AKT1 lacking its PH domain fused to CRY2 together with CIBN equipped with the myristoylation/palmitoylation sequence of the tyrosine kinase FYN. Light-activated AKT1 induced a positive feedback loop on PI3K activity as predicted by computational modeling. Stimulation of AKT by defined temporal light patterns was crucial for stimulating the AKT-FoxO pathway accompanied by the expression of Atrogin-1 involved in muscle atrophy [66]. Deletion of the PH domain was not necessary for light-regulated AKT1, full-length based optoAKT1 even resulted in a slightly elevated kinase activity [66,67]. Recruitment of CRY2-AKT1 and CRY2-AKT2 to plasma membrane microdomains enriched in lipid rafts (m/p-CIBN) or to non-lipid rafts areas (CIBN-CaaX) similarly stimulated AKT signaling with blue light, whereas targeting to other endomembrane failed to activate AKT [16].

### 3.4. Simultaneous Control of Two Pathways by Light

To regulate two pathways simultaneously, several studies combined two blue light-controlled signal transducers. A comparative study of blue light activatable optoPIP3 (CRY2-iSH2/CIBN-CaaX) and optoAKT2 (CRY2-AKT2/CIBN-CaaX) in 3T3-L1 adipocytes revealed that AKT2 was not as active as optoPIP3 in the transport of GLUT4 storage vesicles to the plasma membrane indicating that PI3K activated signals in addition to AKT2 [72]. Acute injuries generate excessive levels of reactive oxygen species (ROS) leading to cell death. The best characterized pro-survival pathways against oxidative stress are the RAF/ERK and the PI3K/AKT pathways. Accordingly, optoCRAF (CRY2-CRAF/CIBN-CaaX) and optoAKT1 (CRY2-AKT1/CIBN-CaaX) protected against hydrogen peroxide (H_2_O_2_) and other oxidative stressors. While optoCRAF stimulated before exposure to oxidative agents conferred preconditioning protection for oxidative stress, optoAKT mediated postconditioning protection. This indicates that precise timing is a prerequisite for protection by pro-survival pathways [73]. Neural regeneration is regulated by neurotrophin signaling stimulating the RAF/ERK and the PI3K/AKT pathway. OptoRAF but not optoAKT induced neurite outgrowth in PC12 cells and sensory neuron dendrite branching in *Drosophila* larvae. Activation of optoRAF and optoAKT promoted axon regeneration in the peripheral and central nervous system in *Drosophila* indicating that these two subcircuits share some downstream components in neuroregeneration [74].

Recently, multichromatics emerged as a powerful approach to control intracellular signaling. By combining two or more differently colored optogenetic tools, each signaling protein can be controlled orthogonally. Kramer et al. combined CRY2/CIBN-based optoAKT and PHYB/PIF-based optoSOS [5]. Red light activated optoSOS but did not affect optoAKT. However, PHYB could be cross-activated by blue light [75]. To solely stimulate optoAKT and keep optoSOS in its inactive state, blue light and far-red light could be applied simultaneously. Alternating pulses of blue/far-red light with pulses of blue/red light kept optoAKT in its active state while optoSOS repeatedly switched between the active and inactive state [5]. Similarly, Bugaj and Lim combined PHYB/PIF-based optoSOS and iLID-based optoPI3K [68,76]. Multichromatic approaches enable the analysis of interconnected signaling pathways in a highly dynamic manner to better understand the temporal aspects of the cross-interaction.

In a further approach, the control of AKT and RAF/ERK signaling was achieved by the scaffold protein connector enhancer of the kinase suppressor of RAS-1 (CNK1). AKT phosphorylation-dependent oligomerization and the subcellular localization control the function of CNK1 [77,78]. In stimulated cells, CNK1 connected AKT and RAF and mediated AKT-dependent phosphorylation and inhibition of RAF [68]. Depending on the cluster size induced by blue light, CNK1-CRY2 (optoCNK1) stimulated RAF/ERK in case of low light intensity and AKT by higher light intensity. OptoCNK1-stimulated AKT counteracted ERK signaling because of the negative feedback of AKT on RAF. In MCF7 cells, optoCNK1 induced ERK-mediated proliferation or AKT-induced differentiation dependent on the light intensity applied to facilitate the control of cell fate decision at the level of AKT and ERK by a single light-controlled signaling protein [68].

## 4. Light Control of Integrin Signaling

Integrins are αβ heterodimeric transmembrane receptors comprising a large extracellular ligand-binding domain and a short intracellular domain [79]. Interaction with the extracellular matrix induces a conformational change in the receptor proteins switching them from a low-affinity to a high-affinity binding state towards their effector proteins. Integrin signaling mediates cell adhesion and migration as well as wound healing and cancer metastasis (Figure 4).

Liao et al. focused on controlling the integrin αVβ3 by utilizing the switch TULIP (Figure 4A) [80]. Therefore, the cytoplasmic tail of β3 was fused to AsLOV2pep and the intracellular adapter protein kindlin-2 to ePDZ. Upon illumination, αVβ3 and kindlin-2 colocalized in focal adhesions and at the cellular periphery, promoting endothelial cell migration, podosome formation, and angiogenic sprouting. Mutational analysis indicated that apart from its binding to αVβ3, kindlin-2 interacts with further signaling proteins such as the integrin-linked kinase and the tyrosine kinase SRC.

Clustering of stimulated integrins recruits the focal adhesion kinase (FAK) to integrin signaling complexes. Subsequently, a conformational change releases the intramolecular inhibition of FAK monomers facilitating dimerization, transphosphorylation, and activation of its kinase activity. Full stimulation of FAK activity involves an SRC-dependent positive feedback loop. OptoFAK represents a single component, blue light-activatable FAK based on FAK-CRY2 oligomerization (Figure 4B) [83]. Light-induced FAK oligomers recruited to focal adhesion sites and mediated phosphorylation of SRC, paxillin, and p130CAS, components of the integrin signaling.

Physical deformation of the plasma membrane generates mechanical forces that modulate cell behaviors like migration. To mimic this process, an optogenetic approach based on the switch iLID was applied to accumulate high levels of a fluorescent protein to the plasma membrane inducing local inward membrane curvature and tension decrease (Figure 4C) [82]. Light-controlled perturbation activated integrin-dependent stimulation coupled to SRC and ERK signaling and cell migration.

To directly control the activity of integrins, the interaction between integrins and the extracellular matrix acts as a model. OptoIntegrin comprises β3 carrying an insertion of PIF in its extracellular domain co-expressed with αV wild type and a matrix functionalized with PHYB (optoMatrix) (Figure 4D) [81]. Upon exposure with 660 nm light, the binding of the optoMatrix to cells expressing optoIntegrin stimulated integrin signaling including ERK. This activation was reversible by 740 nm light. This approach allowed the temporal and spatial control of cell adhesion and cell spreading.

Integrin-mediated cell adhesion is based on its interaction with specific motifs in proteins of the cellular matrix such as the arginine-glycine-aspartic acid (RGD) motif. For engineering, a photoswitchable synthetic ECM protein, opto-ECM, the RGD cell adhesion peptide was inserted in the Jα-helix of LOV2 resulting in a hidden RGD motif in the dark and its accessibility in blue light [84]. Thus, a surface coated with photo-ECM enabled adherence of cells by binding of endogenous integrin to RGD motifs controlled by blue light. To this blue light opto-ECM, a green light-controlled ECM-cell adhesion system was added in a further study [85]. The protein CarH from the switch TtCBD, carrying the cofactor AdoB12 to sense green light, forms tetramers in the dark and dissociates into monomers in green light. The tool GREEN-ON is based on an RGD-coated surface that is masked by a nonadhesive layer of CarH tetramers in the dark preventing binding of cells through integrins. Green light dissociated and removed CarH complexes facilitating binding of integrins to the RGD matrix. The tool GREEN-OFF consists of RGD motifs fused to CarH. Tetrameric CarH-RGD bound to the surface in the dark. RGD was exposed and cells could bind by their integrins to the surface. Green light dissociated the tetrameric CarH-RGD leading to the detachment of the cells.

So far, these extracellular optogenetic approaches to control the interaction of endogenous to RGD functionalized surfaces are not applied for studying integrin signaling.

## 5. Optogenetic Control of T Cell Signaling

The binding of foreign peptide-major histocompatibility complexes (pMHCs) to the T cell receptor (TCR) stimulates T cells [86]. Self- and foreign-pMHCs exist, and only the binding of the latter one results in the activation of intracellular signaling pathways that lead to a functional response of the T cell. One important model, which aims to explain how TCR accomplishes the distinction between self- and foreign-pMHCs is the kinetic proofreading model. It predicts that differences in the binding kinetics determine whether downstream signaling is induced or not.

Two groups independently developed an optogenetic approach to test the kinetic proofreading model. The optogenetic tool opto-ligand-TCR consists of PIF that was inserted into the extracellular domain of a secretion-optimized TCRβ and a tetramerized PHYB as a ligand (Figure 5A) [87]. Yousefi et al. deployed the optogenetic switch PHYB/PIF6 because of the unique property of PHYB to switch constantly between the PIF binding and non-binding state under continuous red light exposure. Higher light intensities resulted in a higher switching rate and darkness, the respective state was stable for hours. Thus, the binding half-life of the TCR-PIF-PHYB interaction could be controlled by the red light intensity. Accordingly, a lower light intensity, which resulted in a longer binding half-life, was required to induce T cell signaling, whereas higher light intensities and therefore shorter binding half-lives were unable to activate T cells. Mathematical modeling of the experimental data provided further evidence for the kinetic proofreading model [87].

The light-controlled chimeric antigen receptor (CAR) is based on the switch LOVTRAP [88]. AsLOV2, presented on a supported lipid bilayer, served as a ligand, and CAR was fused to the AsLOV2 binding partner Zdk. CAR-Zdk bound to the Jα-helix of AsLOV2 in the dark inducing T cell signaling, whereas Zdk dissociated from AsLOV2 under blue light exposure within seconds (Figure 5B). Higher light intensities led to a shorter binding half-life and lower intensities led to a longer binding half-life. The kinetic proof-reading model was tested by exposing the system to different light intensities while using different concentrations of AsLOV2. By relying on the interaction of an optogenetic switch, other biophysical parameters such as the mechanical stability of the TCR-ligand interaction were not changed. Thus, the binding half-life rather than the receptor occupancy was the determinant for CAR signaling [88].

Apart from the TCR-ligand binding half-life, temporal stimulation patterns play a major role in T cell activation. In the case of the iLID-based CAR, CAR was split to separate the ligand-recognition and the signaling domain. Blue light illumination induced dimerization of the split CAR through the interaction of AsLOV2-SsrA with SspB and, in combination with antigen stimulation, activated CAR signaling. Using gene expression as readout, optoCAR was exposed to various light oscillation durations. The results showed that T cells could temporally filter minute-scale signals via a band-stop mechanism, besides second-scale TCR-ligand interactions according to the kinetic proofreading model [89].

Another optogenetic approach to activate T cell signaling utilized the clustering capability of CRY2. The ζ-chain of TCR was fused to CRY2 and attached to the plasma membrane by the myristoylation sequence of the tyrosine kinase LCK. Illumination with blue light led to the clustering of the ζ-chain and subsequent activation of the downstream cascade and initiated Ca^2+^ influx [90].

## 6. Control of Subcellular Localization and Spatial Resolution of Signaling Processes

An outstanding feature of optogenetics is its ability to be applied in a high spatial resolution. The spatial resolution allows the control of single cells in a cell population and even designated subcellular areas inside of the cell, like a certain organelle or areas of the plasma membrane (Figure 6).

A possible application for the high spatial control of optogenetics is the targeting of subcellular compartments of the cell. For an in-depth review of optogenetic control of cell dynamics and subcellular applications see the recently published reviews [91,92], An example is the optogenetic TULIP system based on the LOV2-domain that was applied for local activation and recruitment of the RhoA GTPase to the membrane in epithelial cells (Figure 6A). This optogenetic tool enables a short and spatially controlled activation of RhoA. Short local recruitment of RhoA did not disturb the integrity of the tissue shape homeostasis of the epithelial [93]. On the other hand, it was also possible to influence whole areas of the cell to force a polarization of the cell and its signaling. Another optogenetic system utilized the RhoA GTPase fused to the photoreceptor BcLOV4 derived from a flavoprotein of *Botrytis cinereal*. RhoA-BcLOV4 translocated to the plasma membrane forced a blue light-dependent contraction of the cell and induced Rho-dependent mechanotransduction [94]. By changing the used optogenetic system one can achieve a greater understanding of Rho signaling in different spatially controllable resolutions to either a small subcellular fraction of the cell or the polarization of one side of the cell.

Most studies are focusing on the activity of kinases rather than phosphatases, however, there are a few examples for the optogenetic control of phosphatases. For light control of the protein tyrosine phosphatase 1B (PTP1B), PTP1B was fused to the N-terminus of AsLOV2. OptoPTP1B was constitutively active in the dark. Exposure to blue light caused an allosteric conformational change in the fusion protein that inhibited PTP1B phosphatase activity. Using a Förster resonance energy transfer (FRET)-based readout, a local change in activity in a circular area of about 5 µm shows that dephosphorylation could be spatially controlled in a small area of the cell without direct influence on the rest of the cell [95]. Additional to spatial control of an area of the cell or the membrane was the recruitment of signaling molecules to the membrane to bring it into proximity of e.g., interaction partners.

Spatially controllable PIP3 production on the plasma membrane using CRY2 fused with a constitutively active PI3-kinase and a plasma membrane-anchored CIBN system was applied to regulate blue light-induced movement towards or away from the signaling source by the creation of a PIP3 gradient [96] (Figure 6B). This could be an alternative to induction with second messengers because the dosage and spatial control is unique and enables a clearer picture of the downstream signaling networks. Second messengers are not only being produced by light but also via control of Ca^2+^ ions. By anchoring CRY2-STIM1 (stromal interaction molecule 1) to ER-anchored CIBN and inducing oligomerization of CRY2-STIM1, local puncta-like structures were generated going along with Ca^2+^ influx and the subsequent signaling events [97]. Spatial control on an even smaller scale could be achieved by specifically targeting organelles and achieving organelle-specific effects. The cAMP-dependent protein kinase (PKA) was sequestered at different subcellular locations and has site-specific effects on the cell. To utilize this in a spatially controllable fashion, an optogenetic PKA was created using CRY2-PKA and different CIBN anchors. OptoPKA recruited to mitochondria (Figure 6A) and the plasma membrane showed organelle-selective phosphorylation of substrates and thus far unknown substrate targets were found by proteomic analysis, displaying how the spatial control of optogenetics can help to understand the organelle-specific functions of important signaling proteins [98]. To take this another step further, an optoToolbox was created to target a multitude of the target inside of the cells. The system optogenetic endomembrane targeting (OpEn-Tag) enabled the control of the subcellular localization of a protein by light. This toolbox facilitated the recruitment of a protein of interest to the cytosol, defined regions of the plasma membrane by a CaaX or m/p-anchor, focal adhesions, the endoplasmic reticulum, the outer mitochondrial membrane, the Golgi apparatus, and the inner nuclear membrane. As a proof of principle for a context-dependent activity, both CIBN-fused plasma membrane anchors recruited and activated CRY2-AKT1 and CRY2-AKT2 whereas other membrane anchors failed to stimulate AKT signaling by blue light. By adjusting the anchor and bait, this tool can be applied to a variety of signaling applications [16].

## 7. Temporal and Signal Strength Control of Signaling Pathways

One of the main advantages of applying optogenetics for the control of intracellular signaling is the unmatched temporal resolution. Unlike chemical inducers, light enables flexible ON-and OFF switching of the stimulating input to mimic the dynamic properties of signaling (Figure 6C). That way, distinct physiological functions can be elucidated that result from specific dynamics of the same signaling pathway.

To this date, a variety of optogenetic tools has been developed to control the RAS/RAF/ERK signaling cascade, as discussed before. Likewise, this pathway has also been the target of many studies aimed at elucidating its temporal dynamics. By utilizing optoSOS, varying illumination patterns, and a live-cell reporter, Toettcher et al. demonstrated that the RAS/RAF/ERK pathway acts as a high bandwidth transmitter. It filtered out very short timescale stimulations but accepted all other input signals up to many hours. Downstream targets of the ERK pathway were able to react to specific RAS/RAF/ERK pathway dynamics, such as STAT3 that responded to continuous ERK activation [57]. In addition, repeated ERK pulses induced by optoSOS led to higher immediate early gene expression than constantly active ERK. Yet, other signaling pathways affected individual immediate early gene response on the protein level, indicating that a combination of dynamic filtering and logic gates was involved in its activity [69]. Bugaj et al. evaluated if cancer mutations affect signaling dynamics. OptoSOS expressed in cancer cells with corrupted signaling dynamics and normal cells were stimulated with different red light pulse frequencies in a multiparallel manner. When the OFF time between the stimulating light pulses was reduced to the minute range, the RAS-ERK cascade of the cancer cells but not the normal cells lost their temporal resolution, leading to continuous ERK activity. Enhanced BRAF dimerization through particular mutations or targeted drugs was identified as the source of the extended signal decay. Therefore, the RAS-ERK cascade lost its dynamic filtering capability, leading to a deregulated signaling interpretation so that non-proliferative input signals were interpreted as proliferative signals [99].

By utilizing CRY2-based clustering of the low-density lipoprotein receptor-related protein 6 (LRP6), Repina et al. demonstrated that Wnt signaling could be controlled by the light intensity applied, thereby mimicking signaling gradients as they typically happen during developmental processes. Furthermore, varying illumination durations up to 45 h were used to activate Wnt-dependent gene expression. The longest illumination duration led to the highest response, indicating that sustained Wnt activity necessary to activate gene expression [4].

Li et al. developed CRY2/CIBN-based optoTGFBR for control of TGF-β signaling. Using Smad2 as a downstream readout, optoTGFBR signaling strength could be fine-tuned by the light intensity applied. Demonstrating the unmatched spatiotemporal resolution of optogenetics, a single short light pulse, continuous high-frequency light pulses, and short light pulses intermitted by long durations in darkness respectively could induce transient, sustained, and oscillating Smad2 activity in single cells (Figure 6D) [7].

## 8. Conclusions and Outlook

Since its first applications, optogenetics evolves rapidly and now provides valuable tools for basic research and applied sciences. In this review, we focused on the applications of optogenetics in controlling signaling pathways in cells by light. Future aspects of optogenetics comprise the improvements of the technology. To increase the throughput, there are multiple approaches to improve the reproducibility and the variety of conditions tested in a single experiment. The amount of time that light is applied to the optogenetic system is as essential for the effectiveness of the system as the light intensity. 96-well opto-plates are a great way to decrease the workload and identify the ideal conditions for each experiment in an efficient manner [76]. A challenge lies in the control of the LEDs as they need to be accessed by control commands. Improvement of the accessibility with included GUI [100] and open-source building plans for 3D prints, including heatsinks, direct this tool into a great asset to researchers applying optogenetics.

An interesting perspective is the combinations of optogenetics with nanobodies to enable light-dependent targeting of endogenous proteins to enable reversible alteration of their activity and function [101]. It is also possible to control apoptosis pathways of the cells intrinsically over optoBax [102]) or extrinsically over Fas and the Fas-associated protein with death domain [103]. These are just some of the recent developments of the use of the optogenetic application in signaling and other reviews give great insight into the wide application field of optogenetics [2].

So far, medical applications of optogenetic tools are rare. There is the problem of the low permeability of the skin by blue light and the resulting question of how to apply directed light to the area of interest. One way to direct the light to the area of interest are fibers that are applied in neurotechnology to guide the light to the desired area. Recently, biodegradable polyethylene glycol diacrylate (PEGDA) linked with DL-dithiothreitol (DTT) could be used to photo induce cell migration and drug release, while being able to have its lifetime adjusted to the desired time frame of the application [104]. To take advantage of the unique abilities of optogenetics there have been different ideas for a possible application like the control of the blood homeostasis in mice [105], targeting cancer cells via light for T-cells [106], gene therapy approaches to tackle breast cancer [107], adenoviral based therapy to selectively target cancer cells with light and reduce off-target effects [108,109]. Combining therapeutic approaches with optogenetic may result in an improvement of the safety and anti-cancer selectivity, making it a promising addition to existing therapeutic approaches to enhance their effectiveness.

With all the advances that were accomplished in the last decade, optogenetics shows its potential in the precise and unique control of cells and their signaling spatially and temporally. The limit of optogenetics have not been reached and the next years will be able to show even more of the potential of this rather new technology.

## Figures and Tables

**Figure 1 ijms-22-05300-f001:**
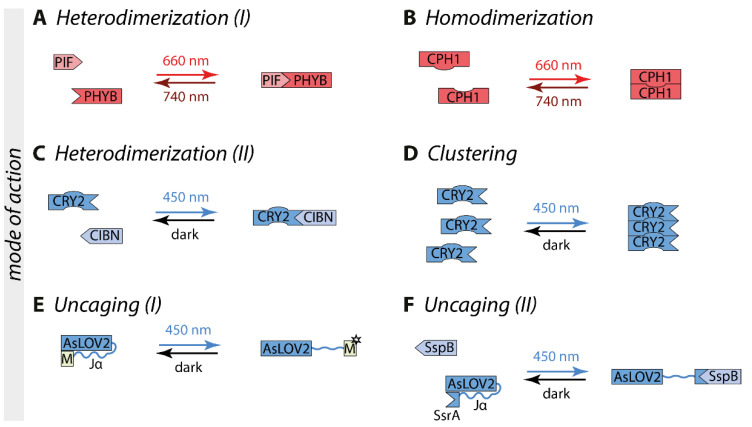
Modes of actions of optogenetic switches are illustrated on commonly used switches. (**A**) PHYB heterodimerizes with its interaction partner PIF upon red light illumination, which can be reversed by far-red light [18]. (**B**) Upon red light exposure, CPH1 forms homodimers that dissociate with far-red light [19]. (**C**) CRY2 engages its interaction partner CIBN to form heterodimers upon exposure with blue light [20]. (**D**) Additionally, CRY2 can form homo-oligomers [21]. (**E**) The Jα helix of AsLOV2 unwinds upon blue light illumination, thereby uncaging a motif of interest (M) [22,23]. (**F**) In the iLID switch, the adapter protein SsrA is exposed under blue light, enabling the binding of its interaction partner SspB [24]. Abbreviations: AsLOV2, the light-oxygen-voltage domain of phototropin 1 derived from *Avena sativa*; CIBN, cryptochrome-interacting basic helix-loop-helix 1 (N-terminal amino acids 1-100); CRY2, cryptochrome 2; M, the motif of interest; PHYB, phytochrome B; PIF, phytochrome interacting factor.

**Figure 2 ijms-22-05300-f002:**
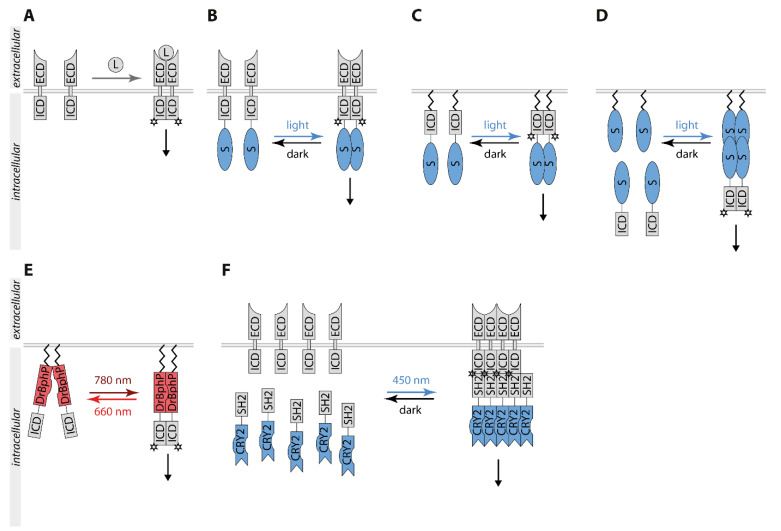
Schematic overview of RTK signaling controlled by optogenetic tools. (**A**). RTKs dimerize upon binding of the ligand (L) to the extracellular domain (ECD), stimulating the kinase activity located in the intracellular domain (ICD) and subsequently RTK downstream signaling. (**B**). The fusion of an optogenetic switch (S) to full-length RTKs can induce light-dependent dimerization and signal independent of ligand binding. (**C**). An optogenetic switch fused to the intracellular domain of RTKs can induce light-dependent dimerization of ICD and RTK signaling. (**D**). An optogenetic switch anchored to the plasma membrane and fused to the intracellular domain of RTKs can induce light-dependent dimerization of ICD at the plasma membrane, activating downstream signaling. (**E**). DrBphP fused to the ICD of RTKs can induce light-dependent dimerization of the ICD, activating downstream signaling. (**F**). Clustering indirectly using CRY2 (CLICR). Though clustering of CRY2 fused to a specific SH2 domain, endogenous RTKs can be targeted and activated independently of ligand stimulation. Abbreviations: CRY2, cryptochrome 2; DrBphP, *Deinococcus radiodurans* bacterial phytochrome; ECD, extracellular domain; ICD, intracellular domain; L, ligand; SH2, Src homology region 2.

**Figure 3 ijms-22-05300-f003:**
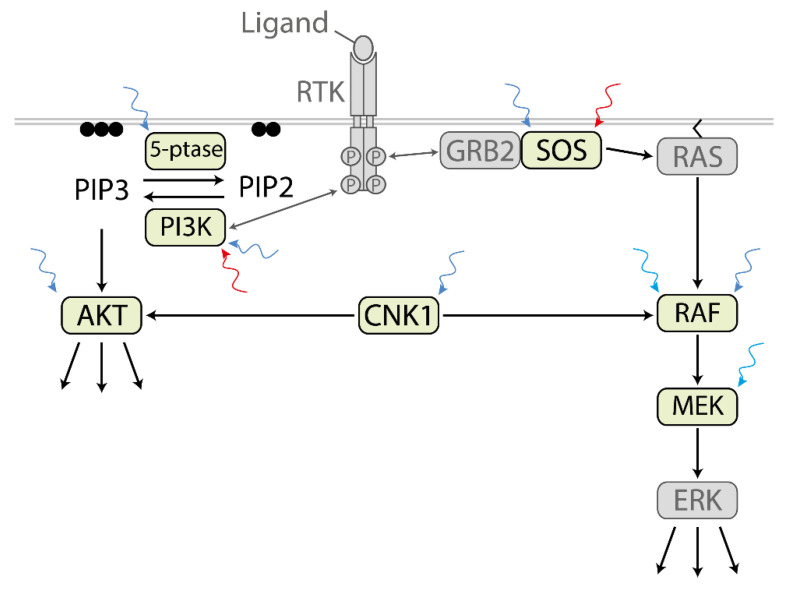
Optogenetic control of PI3K/AKT and MAPK signaling. A wide variety of optogenetic tools is available for control of individual signaling proteins downstream of receptor tyrosine kinases (RTK). Different optogenetic switches were used for the control of the respective signaling protein. SOS: PHYB/PIF [57] and iLID [58]. RAF: CRY2/CRY2 [59], CRY2high [27], CRY2low [27] and CRY2/CIBN [60,61], pdDronpa1 [32]. MEK: pdDronpa1-based psMEK1 [32,62]. 5-ptase: CRY2/CIBN [63]. PI3K: CRY2/CIBN [64], Magnets [29], PYHB/PIF [65]. AKT: CRY2/CIBN [66,67]. CNK1: CRY2/CRY2 [68]. Abbreviations: 5-ptase, inositol 5-phosphatase OCRL; GRB2, growth factor-regulated binding protein 2; PIP2, phosphatidylinositol 4,5-bisphosphate; PIP3, phosphatidylinositol (3,4,5)-trisphosphate; RTK, receptor tyrosine kinase.

**Figure 4 ijms-22-05300-f004:**
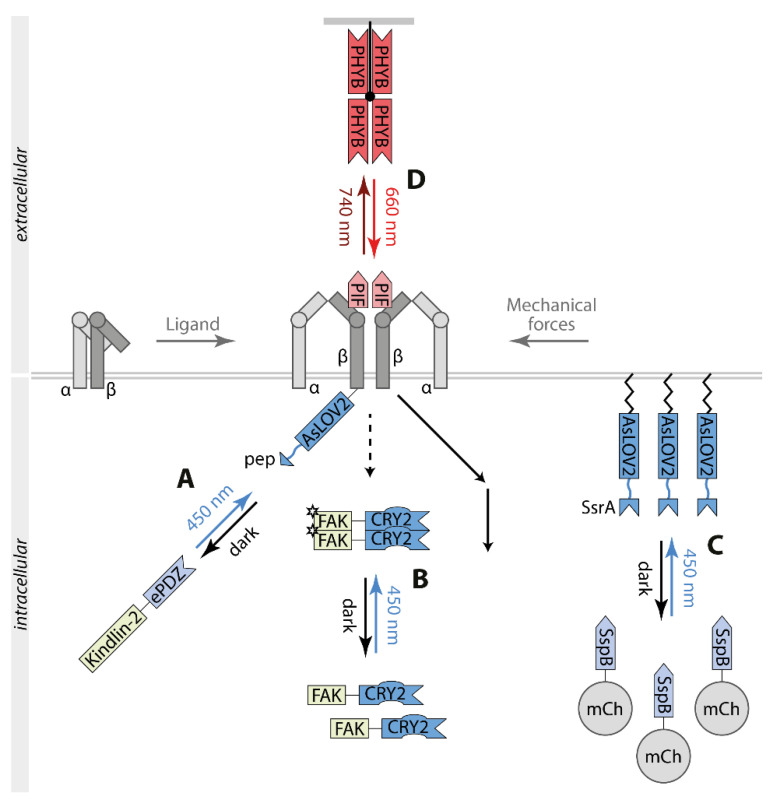
Optogenetic control of integrin signaling. (**A**) Using the TULIP switch the adaptor protein kindlin-2 interacts with the cytoplasmic tail of β3 inducing integrin signaling [80]. (**B**) Homooligomerization of focal adhesion kinase FAK by fusion to CRY2 stimulates signaling at focal adhesion sites [81]. (**C**) The iLID switch allows the recruitment of SspB-mCherry (mCh) to plasma membrane-anchored AsLOV2-SsrA inducing localized curvature and tension decrease correlating with stimulation of integrin signaling [82]. (**D**). PIF inserted in the extracellular domain of β3 mediates integrin clustering by binding matrix-bound PHYB correlating with activation of integrin signaling [81]. Abbreviations: AsLOV2, the light-oxygen-voltage domain of phototropin 1 derived from *Avena sativa*; CRY2, cryptochrome 2; ePDZ, engineered PDZ domain; FAK, focal adhesion kinase; mCh, mCherry; pep, peptide binding to ePDZ; PHYB, phytochrome B; PIF, phytochrome interacting factor; SspB, an interaction partner of SsrA; SsrA, adaptor protein.

**Figure 5 ijms-22-05300-f005:**
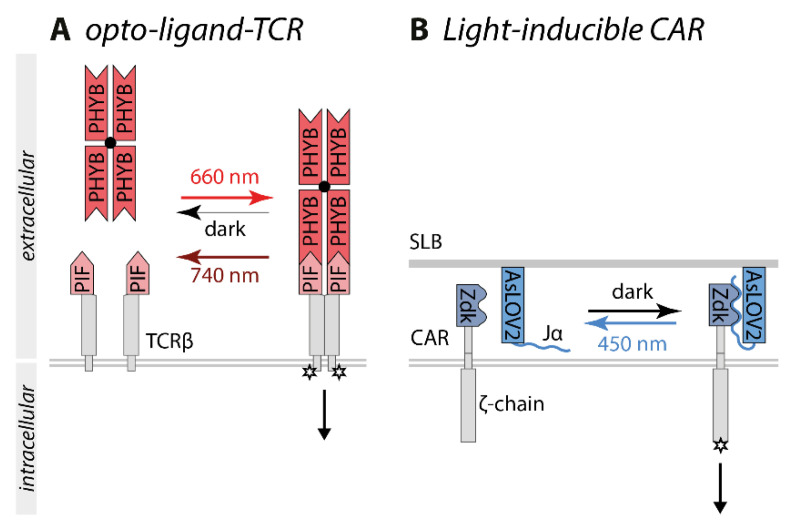
Optogenetic control of T cell signaling. (**A**) In the opto-ligand-TCR system, PIF is inserted into the extracellular domain of TCRβ and tetramerized PHYB serves as ligand. Continuous red light exposure leads to the constant switching between the PIF-binding and non-binding states of PHYB. Higher light intensities induce higher switching rates, enabling the control of the binding half-life by the light intensity applied [87]. (**B**) The LOVTRAP-based Zdk CAR comprises Zdk fused to CAR and AsLOV2, presented on a supported lipid bilayer, as ligand. CAR-Zdk interacts with AsLOV2 stimulating ζ-chain-dependent signaling in the dark. Upon blue light exposure, AsLOV2 and Zdk dissociate correlating with the termination of signaling. The binding half-life can be controlled by the light intensity [88]. Abbreviations: AsLOV2, the light-oxygen-voltage domain of phototropin 1 derived from Avena sativa; CAR, chimeric antigen receptor; PHYB, phytochrome B; PIF, phytochrome interacting factor; SLB, supported lipid bilayer; TCR, T cell receptor.

**Figure 6 ijms-22-05300-f006:**
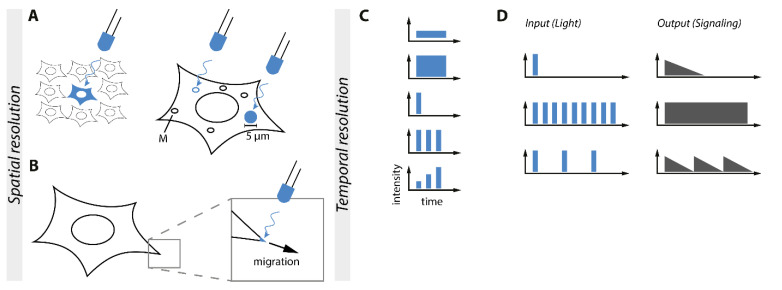
Optogenetics enables unmatched spatiotemporal control of intracellular signaling. (**A**). Schematic overview of the spatial resolution of optogenetics. Single cells or even subcellular sections and compartments of the cell such as mitochondria (M) can be individually addressed. (**B**). Small sections of the membrane can be illuminated to induce migration. (**C**). Schematic overview of the temporal resolution of optogenetics, facilitating complex input patterns. (**D**). Correlation between input and output. Varying input light patterns induce different output signaling patterns generated by the same signaling module as shown for optoTGFBR [7].

**Table 1 ijms-22-05300-t001:** Optogenetic switches for controlling intracellular signaling.

Group	Switch	Co-Factor	Mode of Action	λ (Excitation)	λ (Reversion)	Excitation Time	Reversion Time	Selected Publications
Crypto-chromes	CRY2/CIB1	FAD	heterodimerization	450 nm	dark	seconds	minutes	[20]
	CRY2/CRY2	FAD	homo-oligomerization	450 nm	dark	seconds	minutes	[21]
	CRY2olig	FAD	homo-oligomerization	450 nm	dark	seconds	minutes	[25]
	CRY2clust	FAD	homo-oligomerization	450 nm	dark	seconds	minutes	[26]
	CRY2high	FAD	homo-oligomerization	450 nm	dark	seconds	minutes	[27]
	CRY2low	FAD	heterodimerization	450 nm	dark	seconds	minutes	[27]
LOV domains	AsLOV2	FMN	intramolecular conformational change	450 nm	dark	seconds	tens of seconds	[22,23]
	iLID	FMN	heterodimerization	450 nm	dark	seconds	seconds to minutes	[24]
	LOVTRAP	FMN	heterodimerization, dissociation	450 nm	dark	seconds	seconds to minutes	[28]
	Magnets	FAD	heterodimerization	450 nm	dark	seconds	seconds to hours (variants available)	[29]
	TULIP	FMN	heterodimerization	450 nm	dark	seconds	seconds to minutes	[30]
	VfAU1-LOV	FMN	homodimerization	450 nm	dark	seconds	minutes	[31]
Fluorescent proteins	pdDronpa1	none	homodimerization, dissociation	500 nm	400 nm	seconds	seconds	[32]
Cobalamin-binding domains	MxCBD	CBD	homotetramerization, dissociation	545 nm	dark	n.d.	n.d.	[33]
	TtCBD	CBD	homotetramerization, dissociation	545 nm	dark	n.d.	n.d.	[33]
Phyto-chromes	CPH1	PCB	homodimerization	660 nm	740 nm, dark	milliseconds	milliseconds	[19]
	DrBphP	Biliverdin	homodimerization, dissociation	660 nm	780 nm, dark	n.d.	n.d.	[34]
	PHYB/PIF3 & PHYB/PIF6	PCB	heterodimerization	660 nm	740 nm, dark	milliseconds	milliseconds	[18]

## Data Availability

Not applicable.

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
