# Peer review of "Optogenetic Approaches for the Spatiotemporal Control of Signal Transduction Pathways"

_ijms, 2021, doi:10.3390/ijms22105300_

Round 1
Reviewer 1 Report
Optogenetics is a new technology to optically control various physiological cellular events by genetically encoded photoreceptive proteins. In this article, various non-neuronal optogenetics tools (molecular optogenetics tools) and their applications were reviewed. The authors covered a very wide range of optogenetic switches, optical controlling of several cellular signaling pathways and their applications. Also, the relevant literature is appropriately cited. Since the importance of molecular optogenetics is rapidly increasing in recent years, this review would be of large interest for the researchers to use these versatile techniques to investigate biological events which are being targeted in their researches. Thus, this reviewer basically recommends that the paper should be published in IJMS, but, before doing so, it would be better to reconsider minor points listed below to improve readability.
Minor points
#Since this review include so many abbreviations, it would be useful to include a list of abbreviations into the manuscript, if it is possible.
#Page 2, Line 85
CPH1 first appears here, so that the full-name should also be placed.
#Page 3, Line 107
Should “Figure 1B” be “Figure 1C”?
#Page 6, Line 194 and Page 7, Line 223
“Deinococcus radiodurans”, “Myxoccoccus xanthus”, and “Thermus thermophiles” should be Italicized.
#Page 8, Line 280
The full-name of PKC should be written.
#Page 8, Line 281 and Page 13, Line 536
There are improper spiral symbols.
#Page 8, Line 306
If possible, it would be better to add the information of GRB2 into Fig. 3.
#Page 9, Line 325
“gens” would be a typo of “genes”?
#Page 9, Lines 336 and 342
“rector” (two places) would be a typo of “receptor”?
#Page 10, Line 372
Since “photomimetic mutations” is not a common word, it would be better to more explanation about what mutation is this.
#Page 11, Line 408
What is FYN?
#Page 12, Section 4
It is better to add a map of integrin signaling as a new figure to make the description in this section easier to understand as RTK signaling in Figure 3.
#Page 13, Line 556
What is LCK10?
Reviewer 2 Report
Optogenetic tools allow controlling biological events with light at high spatiotemporal resolution. While they have been developed to control neuronal electrical activity, scientists have also been developing tools to control signal transduction pathways optically. In this manuscript, the authors have provided an overview of optogenetic tools to manipulate signal transduction pathways by focusing on RTK, integrin and T cell signaling pathways. They have also summarized approaches to control signaling pathways in space and time.
The manuscript was written clearly together with informative figures and table. Although the same topic has recently been reviewed elsewhere (e.g., Oh et al., 2021), I strongly believe that the manuscript can still make a positive contribution to the field. By clarifying or addressing the following points, the manuscript will be improved for a publication.
major issues
- GPCRs
Optogenetic control of GPCRs has been explored intensively. Although it is understandable not to cover this large topic, the authors would clarify why this important topic was beyond the scope of this review article.
- “non-neuronal” optogenetic approaches
The term of “non-neuronal” optogenetic approaches is misleading because some approaches have been and can be applied in neurons. The function of neurons is not just about electrical activity. Thus, an alternative term should be considered. For example, “opsin-independent” optogenetic approaches or something more accurate may be appropriate. In addition, “molecular optogenetics” should also be defined clearly.
- A figure for optogenetic control of integrin signaling
Like other sub-topics, providing another figure for integrin signaling will be helpful.
- Neurotechnology
Molecular tools alone are not enough to realize optogenetics fully. Further development of neurotechnology is also essential. Although the current manuscript has briefly mentioned about neurotechnology, the authors should elaborate the discussion about this direction too. See Repina et al. (Cell Reports 2020) and Bugaj et al. (Science 2018).
minor issues
- The subtitle of the section #3 should be updated to reflect the content properly.
- Line 281. “TGF@”.
- Line 323. Insert the citation properly.
- Reference# 107. The format is different from others.
